# Measurement Properties of Self-Report Questionnaires for Amyotrophic Lateral Sclerosis: A Systematic Review and Meta-Analysis of Commonly Used Instruments

**DOI:** 10.3390/ijerph20043310

**Published:** 2023-02-14

**Authors:** Maria Jose Sanchez-Andrades, Maria Jesus Vinolo-Gil, María Jesús Casuso-Holgado, Javier Barón-López, Manuel Rodríguez-Huguet, Rocío Martín-Valero

**Affiliations:** 1Department of Physiotherapy, Faculty of Health Science, Ampliacion de Campus de Teatinos, University of Malaga, C/Arquitecto Francisco Peñalosa 3, 29071 Malaga, Spain; 2Department of Nursing and Physiotherapy, Faculty of Nursing and Physiotherapy, University of Cadiz, 11009 Cadiz, Spain; 3CMU Rehabilitation Intercentres-Interlevels Puerto Real and Cadiz Hospitals, Cádiz-La Janda Health District, 11006 Cadiz, Spain; 4Department of Physiotherapy, Faculty of Nursing, Physiotherapy and Podiatry, University of Seville, 41009 Seville, Spain

**Keywords:** quality of life, amyotrophic lateral sclerosis, design questionnaire, validation studies, patient reported outcome measures, systematic review

## Abstract

(1) Background: Amyotrophic lateral sclerosis (ALS) is a neurodegenerative disease. There is no evidence on the analysis of the measurement instruments available to assess quality of life in these patients, following the consensus-based standards for the selection of health measurement instruments (COSMIN) checklist; (2) Methods: A systematic review was performed in PubMed, Embase, PEDro, Web of Science and Cochrane. The psychometric properties of the questionnaires were determined by using the COSMIN checklist. Two searches were carried out. This systematic review was registered in PROSPERO (CRD42021249005); (3) Results: There were four published articles that analysed the measurement properties in patients with ALS for the following scales: Amyotrophic Lateral Sclerosis Assessment Questionnaire 40, Amyotrophic Lateral Sclerosis-Specific Quality of Life Questionnaire, Short Form 36 Healthy Survey, Epworth Sleepiness Scale and Sickness Impact Profile. Another five scales also met the inclusion criteria: ALS-Depression-Inventory, State Trait Anxiety-Inventory, World Health Organization Quality of Life, Schedule for the Evaluation of Individual Quality of Life, Amyotrophic Lateral Sclerosis Assessment Questionnaire 5. Most Patient Reported Outcome Measures (PROMs) present a low-quality synthesis of evidence. It was observed an excellent pooled reliability of 0.92 (95% Confidence Interval: 0.83–0.96, I^2^ = 87.3%) for four dimensions for questionnaires ALSAQ-40. (4) Conclusions: There is little evidence on generic instruments. Future studies are necessary to develop new tools.

## 1. Introduction

Amyotrophic lateral sclerosis (ALS) is a neurodegenerative disease of the motor neurons of the brain and the spinal cord [1], specifically the primary motor cortex, corticospinal tract, brainstem and spinal cord [2]. The disease progresses rapidly and causes disability in several domains, such as physical mobility or breathing, as well as aspects related to nutrition, communication and emotions [3], and can cause severe cognitive and behavioural impairment [1]. The incidence of ALS is 1.89 per 100,000 inhabitants/year, and is uniform in Western countries, with an average prevalence of 5.2/100,000 [3].

Unfortunately, most people with ALS die within five years [3], mostly due to progressive respiratory muscle weakness, consequent respiratory failure or aspiration pneumonia [1]. There is currently no cure and the only drug available (riluzole) extends life for only a few months (between 3–6) [3], though it leads to useful and visible changes in quality of life (QoL).

A multidisciplinary treatment is the first option for these patients. Changes in diet, speech therapy and non-invasive ventilation are also relevant to extend life [4]. A current systematic review, published in January 2021 [5] shows that therapeutic physical exercise is considered one of the most useful tools to slow the deterioration of the musculature in ALS patients, improving both functionality and pulmonary function [5].

There is a growing interest in researching the quality of life of ALS patients as recent studies have shown that alleviating symptoms, anticipating the progression of muscle weakness, engaging in daily life activities and participation, and ultimately improving quality of life can change the course of the disease and survival [3].

Describing the meaning of quality of life is not easy, as it encompasses several domains of life as well as various individual values. For this review, we define quality of life as a multidimensional construct that includes physical, psychological (anxiety, depression) and even social components, where participation is particularly important, both at home and outside [6].

Due to the lack of evidence, we have found that further research is needed about which questionnaires are better suited to assess this fundamental aspect in patients with neurodegenerative diseases, since recent studies have shown that improving quality of life can affect the course of the disease and survival [3].

## 2. Objectives of this Review

This systematic review aimed to identify all self-report questionnaires on quality of life, sleep quality, fatigue, anxiety and depression for patients with ALS, and to assess the psychometric properties and risk of bias of these questionnaires using the consensus-based standards for the selection of health measurement instruments (COSMIN) checklist.

## 3. Materials and Methods

This systematic review has been performed according to the preferred reporting items for systematic reviews and meta-analyses (PRISMA) checklist [7]. This work is registered in PROSPERO with code CRD42021249005.

### 3.1. Literature Searches and Study Selection

Two search strategies were used for this study: the first was designed to distinguish patient reported instruments and questionnaires (PROMs) used to measure quality of life, fatigue, anxiety, depression and sleep quality in patients with ALS and which of those are used as outcome measures in different research studies (Search A); the second search used a literature review to discover those questionnaires where the measurement properties of the COSMIN questionnaire have been assessed in a sample of patients with ALS (Search B).

SEARCH A: The first search was performed between 14 September and 16 September 2022, in the Cochrane library, Web of Science, PEDro and MEDLINE (PUBMED). The MESH search terms used included “amyotrophic lateral sclerosis”, “quality of life”, “fatigue”, “sleep quality”, “anxiety”, “depression” and “design questionnaire” or “amyotrophic lateral sclerosis” plus a search filter provided by the PROMs group at the University of Oxford [8].

SEARCH B: A search was performed in MEDLINE (PubMed) and Embase between October 2022 and November 2022, using a search filter developed by Terwee [9] to find studies that describe the development of instruments or evaluation of the measurement properties of instruments measuring the quality of life in patients with ALS. No age filters or language restrictions were applied. Search terms included the construct, in this case quality of life, and the population to be studied, i.e., patients with ALS, and the name of the questionnaire found in search A, together with that filter.

A parallel manual search was also carried out, using the same criteria as the previous search; this was used to find studies that describe the development or assessment of the measurement properties of instruments intended to evaluate fatigue, sleep quality, anxiety and depression in ALS patients.

Quality of life is a wide-ranging concept that is affected in a complex way by the physical health of the person, his physiological state, level of independence and social relationships, and the relationship he has with his environment. It includes various aspects, such as fatigue, that is, tiredness that is experienced after an intense and continuous physical or mental effort. Disorder is characterised by extreme tiredness and the inability to function due to lack of energy. Fatigue can be acute or chronic and is also called tiredness. Sleep quality involves both a subjective assessment, as well as quantitative aspects, such as sleep duration, sleep latency or the number of nocturnal awakenings, and purely subjective qualitative aspects, such as the depth of sleep or the capacity to repair it. Anxiety is a normal emotional reaction by the individual to threatening situations. Depression: The WHO defines depression as a frequent mental disorder, characterised by the presence of sadness, loss of interest or pleasure, feelings of guilt or lack of self-esteem, sleep or appetite disorders, feeling tired and lack of concentration [10].

### 3.2. Eligibility Criteria

The inclusion criteria for search A were clinical trials and observational studies, from 2010 to 2022, with ALS patient populations where various PROMs were used to assess some aspect of the quality of life as a measure of the outcome of the study. For search B, the inclusion criteria consisted of articles describing data on the psychometric properties of the scales, without year or language restriction.

The study eligibility criteria for search B had to include these four elements: the self-report must measure some aspect of quality of life; the study sample must be of patients with ALS; and the study objective must be the evaluation of the measurement properties or their development. Studies in which questionnaires were used as outcome measurement instruments, such as clinical trials, were excluded. These studies were discarded in search B, because we attempted to include articles in which the purpose of the study was to analyse the psychometric properties of the outcome measures; hence those that did not have the purpose of explaining the psychometric properties of these scales, as in a clinical trial of a therapy, were discarded.

For the meta-analysis study to be considered as having analysed reliability, the authors had to have calculated at least one of the following coefficients: Cronbach’s alpha (internal consistency), ICC (test-retest reliability), or correlation coefficients comparing the Amyotrophic Lateral Sclerosis Assessment Questionnaire 40 (ALSAQ-40) to other similar scales (parallel-forms reliability). This is in accordance with the consensus-based standards for the selection of health measurement instruments (COSMIN) statement. These criteria were first applied to the title and abstract. All studies which did not meet these criteria were excluded.

The articles describing data on the psychometric properties of the scales, or the scale development were included, since we wanted to assess the measurement properties of each measurement instrument found in clinical trials related to patients with ALS.

### 3.3. Methodological Quality Assessment of the Studies on Psychometric Properties

The consensus-based standards for the selection of health measurement instruments (COSMIN) risk of bias checklist 2018 version [11] was used to assess the risk of bias with respect to different measurement properties, with questions structured in 10 items.

The most important item to assess is the content validity (box 2), i.e., to what extent the content of a PROM properly reflects the construct to be measured. Three aspects of content validity are distinguishable: (a) relevance (all items in a PROM should be relevant for the construct of interest within a specific population and context of use), (b) comprehensiveness (no key aspects of the construct should be missing), and (c) comprehensibility (the items should be understood by patients as intended) [12]. Studies that show information on content validity within the study population should be considered, but the initial PROM development documents and the content itself are also important considerations [13]. Therefore, COSMIN recommends that the quality of a PROM development (box 1) is assessed before the quality of any content validity study, and to assess the content validity in the first place. Should this validity be inadequate, then the PROM would not be recommended.

Then structural validity, internal consistency and cross-cultural validity (boxes 3, 4, and 5) are assessed, which shows the internal structure of the PROM. Structural validity means to what extent the scores of a PROM properly reflect the construct to be measured. Internal consistency measures the degree of interrelation among items, and cross-cultural validity or measurement invariance assesses how well the items of a translated instrument properly reflect the original version [13].

Finally, the remaining measurement properties are evaluated [13] as follows: reliability (box 6) is the probability of obtaining the same score an infinite number of times for the same patient; measurement error (box 7) is closely related to the degree of reliability because it is the systematic error of an individual patient’s score; criterion validity (box 8) shows a proper match to a measurement standard or “gold standard” and, therefore, appears in studies comparing a PROM with an accepted gold standard; the hypothesis test for the construct validity (box 9) aims to compare the instrument with another that is not considered a standard of measurement (convergent validity), or measures the differences among different subgroups (discriminant validity); finally the PROM’s responsiveness or sensitivity to detect changes over time (box 10) is assessed, usually including the expected magnitude of the treatment effect [9,13]. This COSMIN checklist can be found in the Appendix A.

### 3.4. Quality of Measurement Properties

After this assessment, a four-point rating is assigned to each study, where the quality of each standard within a COSMIN box can be rated as “very good”, “adequate”, “doubtful” or “inadequate”. Then the lowest rating of any standard in the box is used as the overall quality rating for each study [14]. Research articles that received a poor COSMIN rating were excluded from further analysis.

### 3.5. Overall Quality of Psychometric Properties

Next, the outcome of each study on a measurement property is rated according to the updated criteria of good measurement properties as “sufficient” (+), “insufficient” (−) or “indeterminate” (¿) using the COSMIN guide [14]. The overall rating of the pooled or summarised outcome could also be “sufficient” (+), “insufficient” (−), “inconsistent” (±) or “indeterminate” (¿).

Finally, the quality of evidence will be graded using the modified GRADE method which rates the evidence as high, moderate, low or very low. The GRADE approach considers the risk of bias of the studies (or the quality of the studies); inconsistency (unexplained inconsistency in study results); imprecision (based on sample sizes); and indirect nature (of the evidence, since there is evidence from populations other than the population of interest of the review). It is assumed that the evidence is high, and the score is gradually degraded based on these aspects [14].

The GRADE approach is used to downgrade evidence when there are concerns about its quality of the evidence. The starting point is always the assumption that the pooled or overall result is of high quality. The quality of evidence is subsequently downgraded by one or two levels per factor to moderate, low or very low evidence when there is risk of bias, (unexplained) inconsistency, imprecision (low sample size), or indirect results. The quality of evidence can even be downgraded by three levels when the evidence is based on only one inadequate study (i.e., extremely serious risk of bias) [14].

### 3.6. Risk of Bias of Included Studies

The risk of bias regarding reliability was calculated for each study selected using the Review Manager version 5.0 with the Cochrane Collaboration tool [15]. The following types of bias were assessed: design requirements and statistical methods with box 6. Two reviewers (M.J.S.A and R.M.V.) assessed the methodological quality and the risk of bias of the studies. In case of doubt, authors resolved disagreements by consensus and consulted a third author (M.J.C.H.) when necessary.

### 3.7. Statistical Analysis

Fixed effects and random effects models were used to obtain the mean of the reliability coefficients (Cronbach’s alpha, ICC, and Spearman correlation coefficients), weighted by the specific weight of each study. First, the coefficients were transformed to stabilise the variances and approximate the results to a normal distribution (Hakstian and Whalen transformation for Cronbach’s alpha and Spearman correlation coefficients) [16,17]. Heterogeneity was assessed using I^2^ statistic by Higgins and Thompson [15]. In the event of heterogeneity, a random effects model was applied, otherwise a fixed effects model was used [18]. Since at least three studies are required to assess heterogeneity, a fixed effects model was used where only two were available [19]. For all analyses, the “metaphor” package of R (version 2.0.0; R Foundation for Statistical Computing) was used to perform meta-analysis on the internal consistency (Cronbach’s alpha) [15]. The “escalc()” function package in R (version 0.5.8; R Foundation for Statistical Computing) was used for meta-analysis on the criterion validity (sensitivity and specificity) [20].

## 4. Results

### 4.1. Systematic Literature Search

In search A, 321 articles were identified in the selected databases. A total of 303 of those were removed after reading the title and abstract, thus leaving 18 articles for full-text review. A total of 13 articles met the Search A selection criteria. The selection process of the articles included in the review can be seen in more detail in Figure 1. The search A strategy is detailed in Appendix A.

In search B, 896 records were identified; after removing duplicates (excluded after reading the title and abstract) and other articles that did not contain a sample of patients with ALS, did not measure properties, or are functional scales, 38 were retained and the others discarded. The search B strategy is detailed in Appendix A.

Of the 38 articles remaining, 18 described the development of each of the scales found in search A (2 scales are described by manuals: ADI-12 and STAI [21,22], so we did not include these), regardless of whether they were validated for the target population or not. After reading the 18 articles on the development of the scale, 10 scales were eliminated, as they did not present patients with ALS in their development, so no search was made for measurement properties for these discarded scales. Of these 8 selected scale development studies, 3 analysed the psychometric properties [23,24,25]. The excluded studies are detailed in Appendix A with reasons.

In addition, 21 studies were found that described the measurement of psychometric properties of these scales in a sample of ALS patients, so the 3 studies mentioned above were included, bringing the total of analysed articles to 24.

### 4.2. Measures of Quality of Life in Amyotrophic Lateral Sclerosis

The ratings were done by a single reviewer. Finally, there were 10 instruments that met the criteria for inclusion, which are as follows: Amyotrophic Lateral Sclerosis Assessment Questionnaire 40 (ALSAQ-40), Amyotrophic Lateral Sclerosis-Specific Quality of Life Questionnaire (ALSSQOL), Short Form 36 Healthy Survey (SF-36), Epworth Sleepiness Scale (ESS), Sickness Impact Profile 68 (SIP 68), ALS-Depression-Inventory and State Trait Anxiety-Inventory, which measure depression and anxiety (the first of these was designed specifically for this purpose); WHOQOL-BREF and SEIQOL questionnaires, which measure the quality of life in a generic way for various populations; and ALSAQ-5.

Table 1 summarises the key characteristics and the COSMIN quality assessment of the development of the PROMs found in search B. A total of 20 specific instruments were found during the searches, with various content: 10 scales measured health-related quality of life (ALSAQ-40, ALSAQ-5, ALSSQOL, EuroQol-5D, McGill Quality of life, PROMIS Global Health, SEIQOL, SF-36, SIP-68 and WHOQOL-BREF) [23,24,25,26,27,28,29,30,31,32]; some did so in more general terms and three of them measured it specifically for ALS patients. Measurable symptoms included pain, physical mobility, psychological aspects, etc. Anxiety and depression were also measured by five instruments (ADI-12, BDI, HADS, HDRS and STAI) [10,33,34,35], but only two were validated for ALS patients, namely ADI-12, with ALS patients included in its development, and STAI. Sleep quality was measured with three scales (PSQI and NHP) [36,37], where the ESS [38] had been tested in ALS patients. Regarding fatigue, the scales used were FSS and CIS-FATIGUE, and none had been validated in people with this disease [39,40]. The COSMIN quality score is not applicable in ADI-12 and STAI, because they are manuals, not articles.

### 4.3. Quality of Measurement Properties

Table 2 shows the quality scores for the studies included in this review corresponding to the final 10 scales where psychometric properties had been assessed with ALS patients. It includes the psychometric property score and the item number of the COSMIN box that served as the basis for this score.

Table 3 provides a summary of the synthesis of evidence and its quality based on the modified GRADE approach [14], the quality of the study and the quality of the psychometric property for the outcome measurement.

### 4.4. Risk of Bias of Included Studies

The Cochrane risk of bias assessment tool [15] was used to assess the risk of bias of the articles included in this review. The COSMIN critical appraisal checklist for reliability (box 6) was applied to the six studies included in the quantitative analysis. The results of the risk of bias regarding reliability in ALSAQ-40 can be observed in Figure 2 and Figure 3. Three of them reported the reliability of the tool ALSAQ-40 [46,47,49] and were assessed as doubtful methodological quality (Figure 2). It should be noted that the risk of bias is adequate only in one of the articles [49] (Figure 2). With respect to statistical methods and overall rating, all were medium risk (Figure 3).

The results of the risk of bias regarding reliability in SF-36 can be observed in Figure 4 and Figure 5. The three studies evaluating the reliability of the tool SF-36 [23,51,52] were rated as having inadequate or doubtful methodological quality (Figure 5). No single study showed a low risk of bias on all domains (Figure 4). The risk of bias was high in two studies for statistical methods (Figure 5). Appendix A shows the questions and answers from box 6 of the COSMIN checklist (reliability) for the studies that analyse this property in ALSAQ-40 and SF-36.

### 4.5. Data Synthesis and Meta-Analysis

To carry out the meta-analysis according to the test-retest reliability assessment, we could only analyse those studies in which the disease-specific ALSAQ-40 questionnaire had been compared with the generic quality of life questionnaire SF-36, as these were the only tests used in more than one group and with the same type of correlation coefficient (Pearson or Spearman). The parallel-forms coefficients could be calculated in all the studies with correlation coefficient (Pearson or Spearman). The restricted maximum-likelihood estimator method was used in this meta-analysis [60], because this method is the default for all frequentist methods in the metafor package. Only three studies were included for reliability analysis [46,47,49].

Regarding evaluation of reliability of the questionnaire ALSAQ-40: Figure 6a shows a reliability analysis for the mobility dimension where we observed an excellent pooled reliability of 0.92 (95% confidence interval (CI): 0.83–0.96, I^2^ = 87.3%). Figure 6b shows a reliability analysis for the ADL dimension; we found an excellent pooled reliability of 0.92 (95% CI: 0.83–0.97, I^2^ = 90.81%). Figure 6c shows a pooled reliability analysis for the eat dimension, where an excellent pooled reliability was also found: 0.93 (95% CI: 0.87–0.97, I^2^ = 84%). Figure 6d shows a reliability analysis for the communication dimension with an excellent reliability of 0.93 (95% CI: 0.85–0.97, I^2^ = 85.5%). Finally, Figure 6e shows a reliability analysis for the emotional dimension, and here pooled reliability was a bit lower at 0.80 (95% CI: 0.66–0.89, I^2^ = 80%).

Figure 7 shows reliability analysis for the generic quality of life questionnaires SF-36 [23,51,52]. Pooled reliability was found to be good 0.75 (95% CI: 0.73–0.77, I^2^ = 0%).

Figure 8 shows internal consistency analysis for the generic quality of life questionnaires SF-36 [23,51,52]. Pooled internal consistency was found to be good 0.86 (95% CI: 0.84–0.87, I^2^ = 25.6%).

Figure 9 shows internal consistency analysis for questionnaire ALSAQ-40. Pooled internal consistency was found to be excellent 0.93 (95% CI: 0.91–0.94, I^2^ = 86%).

## 5. Discussion

In this systematic review we analysed the risk of bias, the findings, and the quality of the existing scientific evidence on studies that assessed the measurement properties of PROMs related to quality of life and sleep, fatigue, anxiety and depression in people with ALS, since these were interrelated aspects of quality of life.

### 5.1. Finding on Psychometric Properties

To our knowledge, this research is the first to provide a comprehensive and systematic overview of the methodological quality of studies and the quality of measurement properties of instruments for quality of life, fatigue, anxiety, depression and sleep in ALS patients, by using the COSMIN checklist [60], which is the only tool capable of methodologically determining the measurement properties of instruments in a standardised way.

Only 10 scales measuring health-related quality of life in the ALS population assessed some of the psychometric properties of the COSMIN questionnaire [24,25,26,33,51,53,54,57,59]. The COSMIN assessment led to heterogeneous results: the content validity studies were all found to be of questionable quality as far as the risk of bias was concerned [25,45,47,50,53]. This is considered the most important property following the COSMIN assessment, and it was only assessed for half of the questionnaires: the disease-specific ALSAQ-40 and ALSSQOL [25,45,47,50], the generic quality of life questionnaires SEIQOL and SF-36, and SIP 68 [53]. This result is due to the lack of assessment of the comprehensibility of the scale when applied to ALS patients. However, in the studies where the property was evaluated, data on the relevance and completeness of the PROM were obtained. Even so, where comprehensibility was neither studied nor mentioned, it was not possible to positively rate it. Inconsistent results could be obtained only for ALSSQOL, as the comprehensibility of the scale was studied through a translated version and was tested in a pilot study. The assessment is a way to encourage and motivate the patient to improve the results; otherwise the patient gradually loses their motivation [24].

Furthermore, the structural validity was only assessed in 50% of the ALS-validated scales (ALSAQ-40, ALSSQOL SF-36 and WHOQOL for quality of life, and STAI for anxiety), whose pooled results show little general evidence and inconsistent results [24,25,50,55,56,57,58]. This happens because many details on how the scale was structurally developed and the statistical methods used were missing, but even in this case, an exploratory factorial analysis was performed for most of them, which was considered optimal and with a proper sample size. However, this was not enough to rate this item as adequate.

All the studies that analysed internal consistency were deemed to be of very good quality, since the only requirement was to use Cronbach’s alpha and reach a score above 0.70. This was assessed for all scales except SIP 68 and WHOQOL, and SEIQOL-DW. This means that all items are related to each other. Cross-cultural validity was assessed on three scales: ALSAQ-40, ALSSQOL, and SF-36 [46,50,56], but the pooled results were inconsistent. It should be noted that only the ALSAQ-40 scale was adapted and validated in Spanish [46]. In addition, there are German, Italian, Turkish, Persian and Portuguese versions [45,47,48,49,61].

The studies that analysed reliability were mostly rated as doubtful for the ALSAQ-40, ALSAQ-5, ALSSQOL, ESS and SF-36 scales [23,46,47,49,51], and their pooled result showed little or very little evidence and inconsistent results. This was due to the inability to establish for certain whether patients were stable from one test to the next, or whether the conditions were the same in both assessments. In the article about the ALSAQ-40 reliability, this test was performed at home, with an average of 2 to 3 months between assessments, and it provided the same data as the ALSAQ-5 scale, since they were analysed in the same article [49]. However, in the article about the same item for the ESS and SF-36 scales [51], the time between reports and scores is unclear, and thus the result is considered doubtful. For SF-36, there are two other articles where this item is unknown or inadequate, and the time between interventions and ratings was 1 month. For ALSSQOL these items were considered appropriate, because the time between assessments was longer, 4 months [50], and there was a calculated intraclass correlation coefficient, which is considered a good reliability model because it measures the degree of agreement between both scores [51]. The pooled result shows moderate evidence with consistent results for this scale.

Reliability may show us the changes in a specific patient, and whether the patient would get the same score an infinite number of times if they were administered the same questionnaire an infinite number of times. This is an important aspect to be able to check differences between patients and is one of the most relevant measures for health professionals when choosing a questionnaire.

On the other hand, the measurement error property was analysed in 3 scales (ALSAQ-40, WHOQOL and STAI) [44,57,58] with variable scores, none of which were inadequate; however, the results are still not consistent for this item.

The criterion validity was not assessed in any quality-of-life questionnaire. It was assessed on an anxiety questionnaire (ADI-12) [33], for which the evidence is moderate and the results consistent, so we can conclude that only this scale is valid for this item. The comparable gold standard was SCID (structured clinical interview) [33].

The studies that analysed the hypothesis test were: 2 inadequate, 6 doubtful and 7 very good [23,24,25,33,46,47,48,49,50,51,58,59]. The heterogeneity of the results is explained by the fact that the scores were determined by the statistical method used to test the hypothesis, considering the Pearson correlation coefficient adequate. However, the *p*-value is not considered optimal [13].

Responsiveness or sensitivity is another fundamental property for this study. It is key when the purpose of the study is the evaluation of the treatment; it was insufficient in the ALSAQ-40 scale [43,44] because it did not provide an adequate description of the intervention that was performed to test the hypothesis. However, ADI 12 and STAI were very good quality studies that were comparable with a gold standard and calculated their sensitivity and specificity [33,58].

Based on the synthesis of evidence and its quality, ALSSQOL had the strongest evidence related to quality; it was the only study to demonstrate content validity, although of doubtful quality, the most important property. It is also one of the instruments with the highest evaluation of psychometric properties. However, it should be considered that the number of studies is limited (n = 2), so more research would be needed to confirm these positive results. Among the generic instruments, data about the WHOQOL instrument are also positive. On the contrary, the ESS and STAI questionnaires show less favourable results, with scarce evidence about their use in patients with amyotrophic lateral sclerosis.

The assessment of the methodological quality of the studies does not imply that the instruments are inadequate. However, it does mean that the reliability and validity of the results obtained with the instrument are questionable.

By means of this meta-analysis, no significant results can be concluded for SF-36 or ALSAQ-40, being the only tools that were used in more than one group and with the same correlation coefficient.

### 5.2. Limitations

Some limitations in our review need to be considered: the search strategy ruled out articles that described all instruments other than quality of life, but since there is controversy and debate about the definition of the quality-of-life concept, this may result in some instruments that describe a patient’s health status rather than their quality of life not being considered for this review. Only articles written in English were included, so language bias could be a problem in this review. In addition to this, the COSMIN approach forced us to assess the studies on measurement properties with a critical eye, but the evidence at hand is sparse and is mostly not reported with the detail needed for this methodology. It entirely depends on subjective judgement, in the same way as the reviewer’s rating of PROMs.

## 6. Conclusions

This review has provided a comprehensive description of the measurement properties of the instruments used to assess quality of life, sleep quality, fatigue, anxiety and depression in clinical trials and different research studies related to ALS patients. Overall, there is little evidence on the more generic instruments. There are more studies on outcome measures that have been specifically developed for these patients, such as ALSAQ-40, ALSSQOL for quality of life, or ADI-12 for depression. As far as the other constructs are concerned, no validated questionnaires with good evidence for use in ALS patients were found.

Although there are many instruments available to evaluate this construct, we recommend developing new tools and studies according to COSMIN standards to evaluate the psychometric properties of these new scales, which is necessary to improve the quality of life in patients with ALS.

## Figures and Tables

**Figure 1 ijerph-20-03310-f001:**
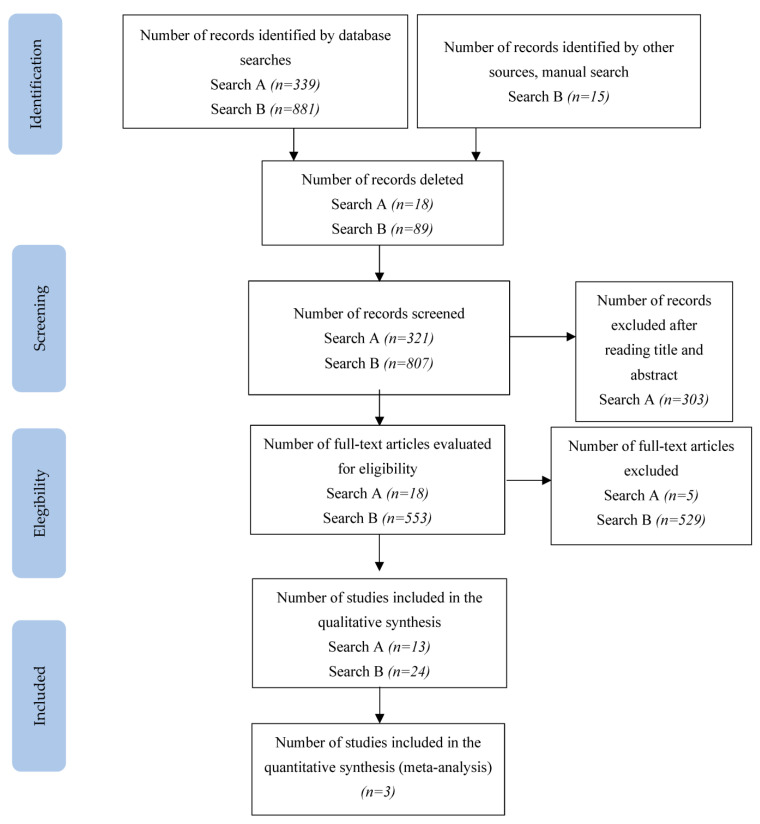
PRISMA flowchart of the study selection process.

**Figure 2 ijerph-20-03310-f002:**
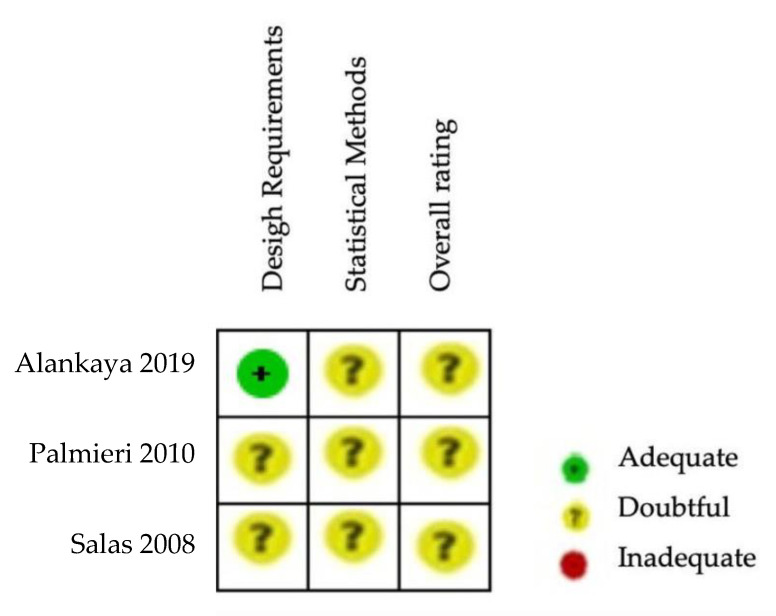
Risk of bias summary for ALSAQ-40 [46,47,49].

**Figure 3 ijerph-20-03310-f003:**
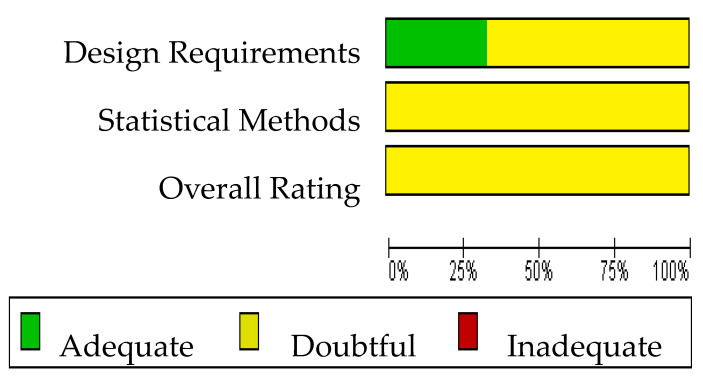
Risk of bias graph for ALSAQ-40.

**Figure 4 ijerph-20-03310-f004:**
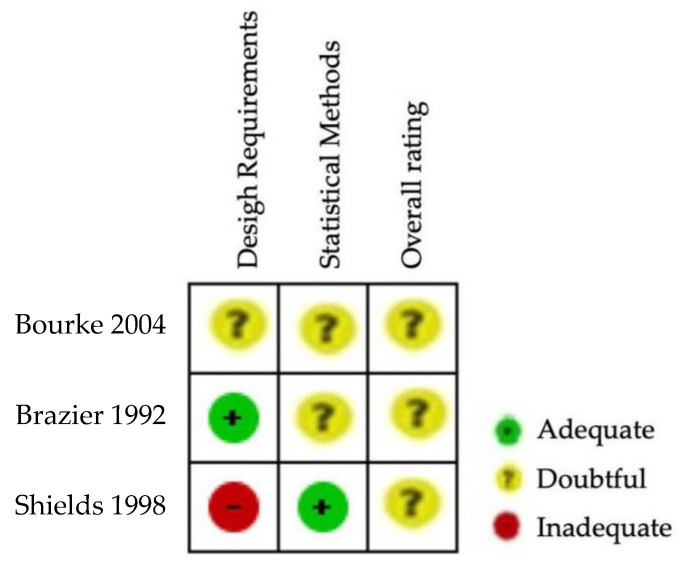
Risk of bias summary for SF-36 [23,51,52].

**Figure 5 ijerph-20-03310-f005:**
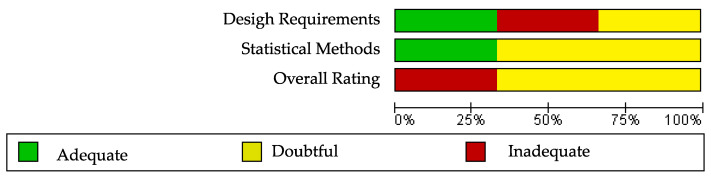
Risk of bias graph for SF-36.

**Figure 6 ijerph-20-03310-f006:**
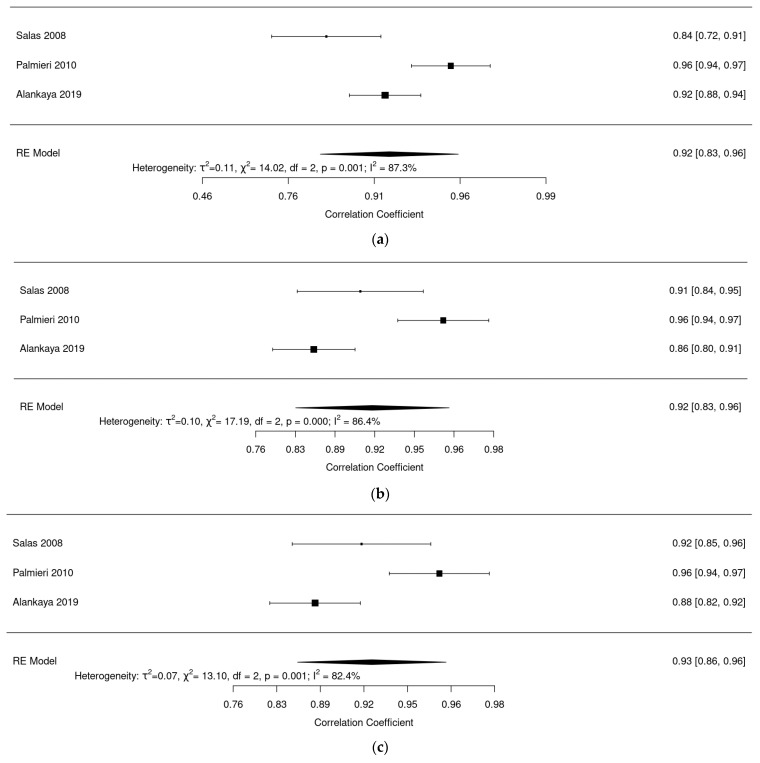
(**a**) Forest plot of mobility dimension for ALSAQ-40. (**b**) Forest plot of ADL dimension for ALSAQ-40. (**c**) Forest plot of eat dimension for ALSAQ-40. (**d**) Forest plot of communication dimension for ALSAQ-40. (**e**) Forest plot of emotional dimension for ALSAQ-40 [46,47,49].

**Figure 7 ijerph-20-03310-f007:**
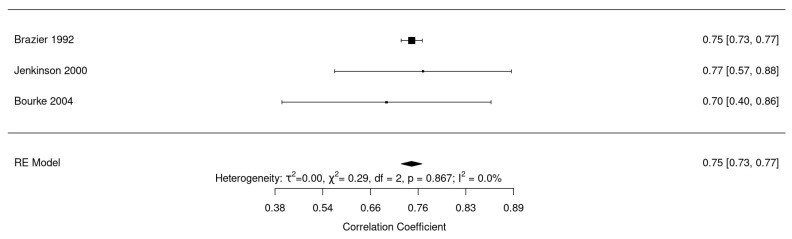
Forest plot for reliability of SF-36 [23,43,51].

**Figure 8 ijerph-20-03310-f008:**
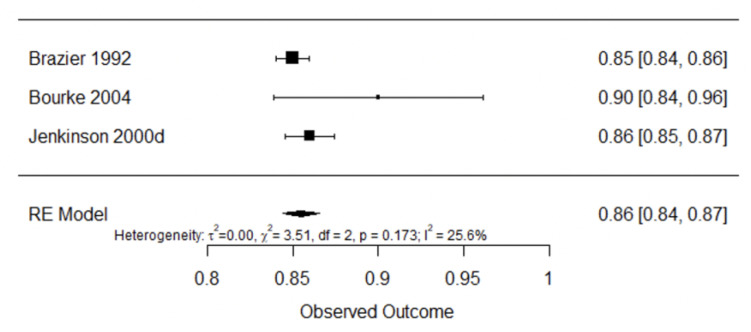
Forest plot for internal consistency of SF-36 [23,43,51].

**Figure 9 ijerph-20-03310-f009:**
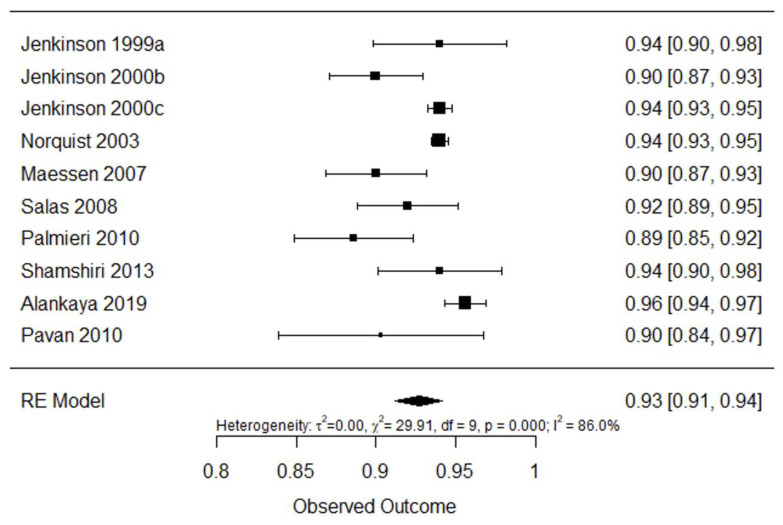
Forest plot for internal consistency of ALSAQ-40 [25,42,43,44,45,46,47,48,49,61].

**Table 1 ijerph-20-03310-t001:** Characteristics and assessment of the quality-of-life instrument development documents included in the review.

Prom, Reference	Construct and Scores	Population	COSMIN Quality Score
ADI-12 Hammer, E.M. [33]	Assess depressive symptoms.	Severely paralysed ALS patients	N/A
ALSAQ-40 Jenkinson, C. [25]	40 items grouped into five dimensions representative of the quality-of-life construct.	Subjective health of ALS patients	Inadequate
ALSAQ-5 Jenkinson, C. [26]	5 grouped items representative of the quality-of-life construct.	Subjective health of ALS patients	Inadequate
ALSSQOL Simmons, Z. [24]	ALS-specific quality of life instrument, consisting of 59 items, scored from 0 to 10.	ALS patients	Inadequate
BDI Beck [34]	Depression. Psychological-cognitive and somatic-vegetative symptoms.	People with mental and behavioural disorders	Inadequate
CIS-FATIGUE Vercoulen, J.H..M.M. [40]	Measures four dimensions of fatigue: fatigue severity, concentration problems, motivation and physical activity.	General population	Inadequate
ESS Johns, M.W. [38]	Designed to measure sleep propensity in a simple and standardised way.	Mental and behavioural illnesses	Inadequate
T.E. Group EUROQOL-5D [27]	Health-related quality of life.	General population and patients with any disease	Inadequate
FSS Krupp, L.B. [39]	Fatigue severity.	Population with disease or healthy population	Inadequate
HADS Zigmond, A.S. [41]	Anxiety and depression. 14 items.	Possible or probable cases of anxiety and depression among patients in nonpsychiatric inpatient clinics	Doubtful
Hamilton Depression Rating Scale Hamilton, M. [35]	Depression. 21 items with the following dimensions: melancholy, somatisation, anxiety and sleep disturbance.	Patients diagnosed with depressive-type disorder	Inadequate
McGill Quality of Life Cohen, S. Robin. [28]	Health-related quality of life.	Persons with life-threatening illness	Inadequate
NHP Hunt, SM. [37]	Energy, pain, emotional reactions, sleep, social isolation and physical mobility.	Generic. General population and patients with any disease	Doubtful
PROMIS v1.1. Global Health Hays, R.D. [29]	Health-related quality of life.	General population and patients with any disease	Inadequate
PSQI Buysse, D.J. [36]	Sleep quality including quantitative aspects of sleep.	Generic, psychiatric and clinical patients	Inadequate
SEIQOL Manual, O’Boyle, C.A. [30]	Health-related quality of life or perceived health. 16 items.	Patients with diseases that influence health status	Doubtful
SF-36 Brazier, J.E. [23]	Health-related quality of life. Eight concepts.	Generic. General population and patients with any disease	Inadequate
SIP-68 Nanda, U. [31]	Using 68 yes or no questions, measures quality of life and level of dysfunction resulting from illness.	General population and patients with any disease	Inadequate
STAI Manual, Spielberg, C. [22]	State and components of trait anxiety.	General population	N/A
WHOQOL-BREF Psychological Medicine [32]	Health-related quality of life.	Generic or any disease	Inadequate

N/A: not applicable. ALS: Amyotrophic Lateral Sclerosis.

**Table 2 ijerph-20-03310-t002:** Quality results of the included studies on self-reported scales.

From	Content Validity	Structural Validity	Internal Consistency	Cross-Cultural Validity/Measurement Error	Reliability	Measurement Error	Criterion Validity	Construct Validity	Responsiveness
ALSAQ-40									
Jenkinson, C. [25]	D (1–31)	I (1)	VG (1,2,5)	N/A	N/A	N/A	N/A	D (3)	N/A
Jenkinson, C. [42]	N/A	N/A	VG (1,2,5)	N/A	N/A	N/A	N/A	N/A	N/A
Jenkinson, C. [43]	N/A	N/A	VG (1,2,5)	N/A	N/A	N/A	N/A	N/A	I (11)
Norquist, J. M. [44]	N/A	N/A	VG (1,2,3,4)	N/A	N/A	A (2)	N/A	N/A	I (11,13)
Maessen, M. [45]	D (15–21)	N/A	VG (1,2,5)	N/A	N/A	N/A	N/A	N/A	N/A
Salas, T. [46]	N/A	N/A	VG (1,2,3,4)	I (3)	D (1,3, 8)	N/A	N/A	D (3)	N/A
Palmieri, A. [47]	D (1–31)	N/A	VG (1,2,5)	N/A	D (1,3,4,8)	N/A	N/A	VG (1,2,3,4)	N/A
Shamshiri, H. [48]	N/A	N/A	VG (1,2,5)	N/A	N/A	N/A	N/A	VG (1,2,3,4)	N/A
Alankaya, N. [49]	N/A	N/A	VG (1,2,3)	N/A	D (4,8)	N/A	N/A	D (3)	N/A
ALSSQOL									
Simmons, Z. [24]	N/A	A (1,3)	VG (1,2,5)	N/A	N/A	N/A	N/A	D (5)	N/A
Oh, J. [50]	D (1–7)	D (4)	VG (1,2,5)	D (1,3)	A (1,3,4)	N/A	N/A	VG (1,2,3,4)	N/A
ESS									
Bourke, S.C. [51]	N/A	N/A	VG (1,2,5)	N/A	D (1,3,8)	N/A	N/A	I (3)	N/A
SF-36									
Brazier, J.E. [23]	N/A	N/A	VG (1,2,3)	N/A	D (3)	N/A	N/A	D (3)	N/A
Shields, R. K. [52]	N/A	N/A	N/A	N/A	I (2)	N/A	N/A	N/A	N/A
Neudert, C. [53]	D (5,6,7)	N/A	N/A	N/A	N/A	N/A	N/A	N/A	N/A
Jenkinson, C. [54]	N/A	N/A	VG (1,2,3)	N/A	N/A	N/A	N/A	N/A	N/A
Bourke, S.C. [51]	N/A	N/A	VG (1,2,3)	N/A	D (1,2,3)	N/A	N/A	I (1)	N/A
Dallmeijer A.J. [55]	N/A	A (1,3)	N/A	N/A	N/A	N/A	N/A	N/A	N/A
Dallmeijer, A.J. [56]	N/A	D (4)	N/A	A (3)	N/A	N/A	N/A	N/A	N/A
SIP-68									
Neudert, C. [53]	D (5,6,7)	N/A	N/A	N/A	N/A	N/A	N/A	N/A	N/A
WHOQOL-BREF									
Young, C.A. [57]	N/A	A (2)	N/A	N/A	N/A	A (3)	N/A	VG (1,2,34)	N/A
ADI-12									
Hammer, E.M. [33]	N/A	N/A	VG (1,3,5)	N/A	N/A	N/A	VG (1,3)	VG (1,2,3)	VG (2,3,4,5,6,7)
STAI									
Siciliano, M. [58]	N/A	A (1)	VG (1,2,5)	N/A	N/A	D (2)	N/A	VG (1,2,3,)	VG (1,2,3)
SEIQOL									
Clarke, S. [10]	N/A	N/A	I (1)	N/A	N/A	N/A	N/A	N/A	N/A
Neudert, C. [53]	Doubtful (5,6,7)	N/A	N/A	N/A	N/A	N/A	N/A	N/A	N/A
Felgoise, S.H. [59]	N/A	N/A	N/A	N/A	N/A	N/A	N/A	VG (1,2,3)	N/A
ALSAQ-5									
Alankaya, N. [49]	N/A	N/A	VG (1,2,3)	N/A	D (4)	N/A	N/A	D (3)	N/A

AD: adequate, D: doubtful, I: inadequate, N/A: not applicable, VG: very good.

**Table 3 ijerph-20-03310-t003:** Synthesis of evidence and quality of evidence.

	Content Validity	Structural Validity	Internal Consistency	Cross-Cultural Validity /Measurement Variance	Reliability	Measurement Error	Criterion Validity	Hypothesis Test for Construct Validity	Sensitivity
ADI-12	N/A	N/A	+/M	N/A	N/A	N/A	+/M	?/L	?/L
ALSAQ-5	N/A	N/A	+/H	N/A	?/VL	N/A	N/A	?/VL	N/A
ALSAQ-40	*R+* *Compl+* *Compr±* M	?/VL	+/H	?/VL	?/L	?/L	N/A	±/L	±/VL
ALSSQOL	*R+* *Compl+* *Compr N/A* M	-/L	+/H	?/L	+/M	N/A	N/A	±/L	N/A
ESS	N/A	N/A	+/L	N/A	-/VL	N/A	N/A	?/VL	N/A
SEIQOL	*R+* *Compl+* *Compr-* L	N/A	-/VL	N/A	N/A	N/A	N/A	?/M	N/A
SF-36	*R+* *Compl+* *Compr-* M	-/L	+/H	?/L	+/M	N/A	N/A	?/VL	N/A
SIP-68	*R+* *Compl+* *Compr±* M	N/A	N/A	N/A	N/A	N/A	N/A	N/A	N/A
STAI	N/A	?/L	?/H	N/A	N/A	?/VL	N/A	?/M	?/M
WHOQOL-BREF	N/A	?/L	N/A	N/A	N/A	+/M	N/A	+/H	N/A

*Compl:* completeness; *Compr*: comprehensibility; H: high; L: low; M: moderate; N/A: not applicable; *R:* relevance; VL: very low; +: sufficient; -: insufficient; ±: inconsistent;?: indeterminate.

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
