# Peer review of "Measurement Properties of Self-Report Questionnaires for Amyotrophic Lateral Sclerosis: A Systematic Review and Meta-Analysis of Commonly Used Instruments"

_ijerph, 2023, doi:10.3390/ijerph20043310_

Round 1

Reviewer 1 Report

Dear Authors..Kindly address the following queries

1. Studies in which questionnaires were used as outcome measurement instruments, such as clinical trials, were excluded. Can you explain the reason?

2. Regarding fatigue, the scales used were FSS and CIS-FATIGUE, and none of them had been validated in people with this disease. Kindly explain the reason behind this?

3. Only 3 studies were included for reliability analysis. May I know the reason behind this?

4. Introduction can be improved further.

5. Sporadic ALS, Familial ALS, Guamanian ALS. Did you included all ALS or any specific ALS?

Author Response

ITEMIZED LIST OF THE REVIEWERS’ COMMENTS

Manuscript ID:  ijerph-2187518

Title: Measurement properties of self-report questionnaires for amyotrophic lateral sclerosis: a systematic review and meta-analysis of commonly used instruments.

Dear Reviewer,

We greatly appreciate the editor´s and reviewers’ kind and encouraging comments about our study. We have followed their suggestions, trying to incorporate them into the revised version of our manuscript. We uploaded the tracked changes manuscript, the clean version revised manuscript and itemized point-by-point response to the reviewer’ comments are presented below.

Editor´s and Reviewers´ comments:

*Reviewer 1

RV: Reviewer

AA: Authors

Dear Authors. Kindly address the following queries

RV: 1. Studies in which questionnaires were used as outcome measurement instruments, such as clinical trials, were excluded. Can you explain the reason?

AA: First, authors want to thank the modifications suggested by the reviewer and his/her effort to improve our manuscript. Following the reviewer´s recommendation, we have attempted to better clarify our eligibility criteria section of the manuscript. The main objective of this systematic review was to assess the psychometric properties and the risk of bias of these questionnaires using the Consensus-based Standards for the selection of health Measurement Instruments (COSMIN) checklist. A new paragraph was added in lines 124 to 128 on page 3 in the revised clean version in the Materials and Methods section. These studies do not provide the necessary information to respond to Objective B of the review. In them, psychometric properties are not studied.

RV: 2. Regarding fatigue, the scales used were FSS and CIS-FATIGUE, and none of them had been validated in people with this disease. Kindly explain the reason behind this?

AA: We consider that these scales are not validated because there are no documents or articles in which this validation is carried out, nor in which the psychometric properties of the scales are measured in patients with ALS. That studies have used non-specific fatigue scales for ALS. It would be a possible prospective study to be proposed.

RV: 3. Only 3 studies were included for reliability analysis. May I know the reason behind this?

AA: We were only able to analyse those studies that included test-retest reliability assessment data for the questionnaire the disease-specific ALSAQ-40. Psychometric properties are qualities of each scale. Therefore, it is only possible to combine in a meta-analysis those studies that have measured the same quality of the same scale. In this case, 3 for ALSAQ-40 and 3 for SF-36.

RV: 4. Introduction can be improved further.

AA: Thank you for your suggestion. We agree with your comment. Following the reviewer´s recommendation, we have added new information. A new paragraph was added in lines 43 to 47 and 53 to 56 on pages 1 and 2 in the revised clean version in the introduction section.

RV: 5. Sporadic ALS, Familial ALS, Guamanian ALS. Did you include all ALS or any specific ALS?

AA: We included all ALS types.

Please, do not hesitate to contact me, if you require further corrections and information.

Thank you in advance

Reviewer 2 Report

Dear authors,

- Line 57 talks about the objectives of the study in section 1. Introduction. The objectives should be explained as their own section after the material and methods.

- In section 2. Material and methods, figure 1 on pages 4 and 5 should appear. A systematic bibliographic search should not be reflected within the results section, since it is a figure that exposes and clarifies the concepts named within the material and methods.

- It would be good to see an annex with the COSMIN consensus, in order to better understand this checklist.

- The tables are very extensive and occupy more than one page. For this reason, it is good, or reduce them in size, or move them to an annex, or occupy 2 pages, but the beginning of the next page must be the header row of the table to remember what each column means.

- The explanation of tables 1, 2 and 3 should appear before the tables and not after. In this way, the table is introduced and the reader knows what is going to be talked about and can pay more attention to those aspects of the table that are most interesting to him.

Author Response

ITEMIZED LIST OF THE REVIEWERS’ COMMENTS

Manuscript ID:  ijerph-2187518

Title: Measurement properties of self-report questionnaires for amyotrophic lateral sclerosis: a systematic review and meta-analysis of commonly used instruments.

Dear Reviewer,

We greatly appreciate the editor´s and reviewers’ kind and encouraging comments about our study. We have followed their suggestions, trying to incorporate them into the revised version of our manuscript. We uploaded the tracked changes manuscript, the clean version revised manuscript and itemized point-by-point response to the reviewer’ comments are presented below.

Editor´s and Reviewers´ comments:

*Reviewer 2

RV: Reviewer

AA: Authors

Dear authors,

RV: 1. Line 57 talks about the objectives of the study in section 1. Introduction. The objectives should be explained as their own section after the material and methods.

AA: Thank you for your suggestion. We have created a new section with the objectives of the study on page 2.

RV: 2.  In section 2. Material and methods, figure 1 on pages 4 and 5 should appear. A systematic bibliographic search should not be reflected within the results section, since it is a figure that exposes and clarifies the concepts named within the material and methods.

AA: Thank you for your suggestion. According to the PRISMA 2020 statement, which presented updated guidance for the publication of systematic reviews, we have considered it appropriate to reflect the flowchart (figure 1) with the results, explaining the selection made

RV: 3.  It would be good to see an annex with the COSMIN consensus, in order to better understand this checklist.

AA: Thank you for your suggestion. We agree with your comment. Following the reviewer´s recommendation, we have added a new annex with the COSMIN consensus.

RV: 4. The tables are very extensive and occupy more than one page. For this reason, it is good, or reduce them in size, or move them to an annex, or occupy 2 pages, but the beginning of the next page must be the header row of the table to remember what each column means.

AA: We agree with your suggestion. Without a doubt, their recommendations help us to give more clarity and structure to the result part. Thank you very much for your comments. We have reduced the size of them.

RV: 5.  The explanation of tables 1, 2 and 3 should appear before the tables and not after. In this way, the table is introduced, and the reader knows what is going to be talked about and can pay more attention to those aspects of the table that are most interesting to him.

AA: Sorry for the inconvenience. Following the reviewer suggestion, we have added before the explanation of tables 1, 2 and 3 in the revised clean version.

Thank you very much.

Please, do not hesitate to contact me, if you require further corrections and information.

Thank you in advance

Reviewer 3 Report

The authors present a systematic review and meta-analysis assessing the on quality of life, sleep quality, fatigue, anxiety and depression of the questionnaires for amyotrophic lateral sclerosis. The approach seems appropriate, the introduction well conducted and supported by pertinent bibliography, and the clinical implications relevant. However, the following aspects should be addressed:

In the selection criteria of the methods debería estar descrito all aspect of the quality of life.

The criteria for the application of the GRADE approach must be carefully explained in the methods section..

Line 88 “A parallel manual search was also performed” match terminology with PRISMA flowchart.

In table 1, the abbreviation ALS must be added at the end of the table and its meaning.

On line 348 amyotrophic lateral sclerosis put abbreviation ALS.

Author Response

ITEMIZED LIST OF THE REVIEWERS COMMENTS

Manuscript ID:  ijerph-2187518

Title: Measurement properties of self-report questionnaires for amyotrophic lateral sclerosis: a systematic review and meta-analysis of commonly used instruments.

Dear Reviewer,

We greatly appreciate the editor´s and reviewers’ kind and encouraging comments about our study. We have followed their suggestions, trying to incorporate them into the revised version of our manuscript. We uploaded the tracked changes manuscript, the clean version revised manuscript and itemized point-by-point response to the reviewer’ comments are presented below.

Editor´s and Reviewers´ comments:

*Reviewer 3

Rv: Reviewer

AA: Authors

The authors present a systematic review and meta-analysis assessing the on quality of life, sleep quality, fatigue, anxiety and depression of the questionnaires for amyotrophic lateral sclerosis. The approach seems appropriate, the introduction well conducted and supported by pertinent bibliography, and the clinical implications relevant. However, the following aspects should be addressed:

RV: 1. In the selection criteria of the methods debería estar descrito all aspect of the quality of life.

AA: First, authors want to thank the modifications suggested by the reviewer and his/her effort to improve our manuscript. We have added a definition of all aspects of quality of life on page 3, lines 101 to 113.

RV:2. The criteria for the application of the GRADE approach must be carefully explained in the methods section

AA: Thank you for your suggestion. We agree with your comment. Following the reviewer´s recommendation, we have added more information about GRADE approach on page 4 and 5, in lines 189 to 201.

RV: 3. Line 88 “A parallel manual search was also performed” match terminology with PRISMA flowchart.

AA: Thank you for your suggestion. We have changed PRISMA flowchart, and we have added the word “manual search” to the identification part

RV: 4. In table 1, the abbreviation ALS must be added at the end of the table and its meaning.

AA: Sorry for the inconvenience. Following the reviewer suggestion, we have added at the end of the table 1 the meaning of the ALS abbreviation.  Thank you very much.

RV: 5. On line 348 amyotrophic lateral sclerosis put abbreviation ALS.

AA: Sorry for the inconvenience. We have added abbreviation ALS now on line 396. With the changes, line 348 is now 396.

Please, do not hesitate to contact me, if you require further corrections and information.

Thank you in advance

Reviewer 4 Report

Authors:

Please complete the following questions as soon as possible!

Q1. Line 28 "Amyotrophic Lateral Sclerosis Assessment Questionnaire 5",please explain clearly.  

Q2. Are there any other methods besides the measurement table mentioned in Line 103-106?

Q3. Can you explain the correct location and content of Line 138-148 Box1-to box10?

Q4. Line 166 Please explain the design method in box6?

Q5. Please modify the arrangement and description of Figure 1 to make it easier for readers to read.

Q6. For Line 244-246, please check whether the correct quantity on the scale is consistent with the quantity mentioned in line 244.

Q7. Please revise the arrangement of Table2. You can provide a header on each page to make it easier for readers to read.

Q8. Table2. There is an "A" symbol displayed, but the author did not clearly explain its meaning.

Q9. Table3. The author of the last two symbols of "ALSAQ-40" did not explain their meaning clearly.

Q10. The author of Line 302-341 confirmed the P value through statistical methods, but did not provide the mathematical model equation. Please add and explain it.

Thank you for organizing these statistical evaluation scales for the clinical performance of ALS patients and making it easier for medical staff to formulate various medical plans, but your suggestions can be applied for clinical trials in the future.

Author Response

ITEMIZED LIST OF THE REVIEWERS’ COMMENTS

Manuscript ID:  ijerph-2187518

Title: Measurement properties of self-report questionnaires for amyotrophic lateral sclerosis: a systematic review and meta-analysis of commonly used instruments.

Dear Reviewer,

We greatly appreciate the editor´s and reviewers’ kind and encouraging comments about our study. We have followed their suggestions, trying to incorporate them into the revised version of our manuscript. We uploaded the tracked changes manuscript, the clean version revised manuscript and itemized point-by-point response to the reviewer’ comments are presented below.

Editor´s and Reviewers´ comments:

*Reviewer 4

Please complete the following questions as soon as possible!

RV: Reviewer

AA:  Authors

RV: 1. Q1. Line 28 "Amyotrophic Lateral Sclerosis Assessment Questionnaire 5”, please explain clearly.  

AA: First, authors want to thank the modifications suggested by the reviewer and his/her effort to improve our manuscript. From lines 23 to 28, I am listing the names of the questionnaires that have been reviewed in the study. Among them is the Amyotrophic Lateral Sclerosis Assessment Questionnaire 5. It is a questionnaire that consists of 5 evaluable items, that is why it is called that. Thank you very much.

RV: 2. Q2. Are there any other methods besides the measurement table mentioned in Line 103-106?.

AA: No, there are no other methods. Primary documents that we have analyzed have not allowed us to do another type of measurement analysis. There are other methods but we have not carried them out.

RV: 3. Q3. Can you explain the correct location and content of Line 138-148 Box1-to box10?

AA: Sorry, we had not mentioned box 1 and box 2. Box 1 deals with analyzing the quality of the development of the articles, carried out in Table 1, and box 2 deals with the content validity. These lines have been modified, and we have also added an annex with the checklists: box 1 to box 10.

RV: 4. Q4. Line 166 Please explain the design method in box6?

AA: Reliability refers to the proportion of the total variance in the measurements which is due to 'true' differences between patients. The word ‘true’ must be seen in the context of CTT, which states that any observation is composed of two components – a true score and error associated with the observation. ‘True’ is the average score that would be obtained if the scale was administered an infinite number of times to the same person. It refers only to the consistency of the score, and not to its accuracy. Reliability can also be explained as the ability of a PROM to distinguish between patients. Within a homogeneous group, it is hard to distinguish between patients. An important assumption made in a reliability study and in a study on measurement error is that patients are stable on the construct to be measured between the repeated measurements. In annex 1, we have added the Box 6 complete, with the design method, and in Supplemntary table S6.

RV: 5. Q5. Please modify the arrangement and description of Figure 1 to make it easier for readers to read.

AA: Thank you for your suggestion. We have removed the reasons for exclusion so there is less content, and it is easier to read.

RV: 6. Q6. For Line 244-246, please check whether the correct quantity on the scale is consistent with the quantity mentioned in line 244.

AA: Thank you for your comment. the error in line 244 is already eliminated. There is the correct quantity. Now, the lines are 284 and 285.

RV: 7. Q7. Please revise the arrangement of Table2. You can provide a header on each page to make it easier for readers to read.

AA: Thank you for your suggestion. We agree with your comment. We have reduced the size of the table to one page to make it easier to read.

RV: 8. Q8. Table2. There is an "A" symbol displayed, but the author did not clearly explain its meaning.

AA: We think the symbol is N/A: not aplicable.

RV: 9. Q9. Table3. The author of the last two symbols of "ALSAQ-40" did not explain their meaning clearly.

AA:  Sorry for the inconvenience. We have added the description on table 3, of the symbols that appear in the table, among which is the one previously mentioned.

RV: 10. Q10. The author of Line 302-341 confirmed the P value through statistical methods but did not provide the mathematical model equation. Prease add and explain it

AA: We agree with your suggestion. Without a doubt, their recommendations help us to give more clarity and structure to the result part. Thank you very much for your comments. We have remade all figures in the revised clean version. We have also improved the description in Figures 6, 7 and 8 in the revised clean version on pages 12, 13 and 14. Regarding statistical method, we have added a new reference in line 353 on page 11 about the method used, which is the "Restricted Maximum-Likelihood estimator". It is the one that performs best, and it is the default estimator in metafor.

New Reference: https://www.ncbi.nlm.nih.gov/pmc/articles/PMC4950030/

Please, do not hesitate to contact me, if you require further corrections and information.

Thank you in advance
